# A Survey of Cannabis Use among Patients with Inflammatory Bowel Disease (IBD)

**DOI:** 10.3390/ijerph20065129

**Published:** 2023-03-15

**Authors:** Alondra Velez-Santiago, Edwin Alvarez-Torres, Ricardo Martinez-Rodriguez, Emmanuel Candal-Rivera, Luis Muniz-Camacho, Luis Ramos-Burgos, Esther A. Torres

**Affiliations:** 1Department of Medicine, School of Medicine, University of Puerto Rico Medical Sciences Campus, PR Medical Center, San Juan, PR 00935, USA; alondra.velez@upr.edu (A.V.-S.); edwin.alvarez@upr.edu (E.A.-T.); ricardo.martinez15@upr.edu (R.M.-R.); 2Veterans Affairs Caribbean Health System, 10 Calle Casia, San Juan, PR 00921-3201, USA; emmanuel.candal@upr.edu (E.C.-R.); luis.muniz.camacho@gmail.com (L.M.-C.); 3Massachussetts General Hospital, 55 Fruit St., Boston, MA 02114, USA; lramosburgos@gmail.com

**Keywords:** inflammatory bowel disease, cannabis, medical marijuana

## Abstract

Inflammatory bowel diseases (IBDs) are chronic conditions of unknown cause or cure. Treatment seeks to reduce symptoms and induce and maintain remission. Many patients have turned to alternatives, such as cannabis, to alleviate living with IBD. This study reports the demographics, prevalence, and perception on cannabis use of patients attending an IBD clinic. Patients agreed to participate and completed an anonymous survey during their visit or online. Descriptive analysis, Fisher’s exact test, and Wilcoxon-Mann-Whitney rank-sum test were used. One hundred and sixty-two adults (85 males, 77 with CD) completed the survey. Sixty (37%) reported use of cannabis, of which 38 (63%) used it to relieve their IBD. A value of 77% reported low to moderate knowledge about cannabis, and 15% reported little to no knowledge. Among cannabis users, 48% had discussed use with their physician, but 88% said they would feel comfortable discussing medical cannabis for IBD. Most saw improvement of their symptoms (85.7%). A considerable number of patients with IBD use medical cannabis for their disease, unknown to their physician. The study reinforces the importance that physicians understand the role of cannabis in the treatment of IBD in order to appropriately counsel patients.

## 1. Introduction

Inflammatory bowel diseases (IBDs) are chronic inflammatory conditions of unknown cause, consisting mainly of ulcerative colitis (UC) and Crohn’s disease (CD). Among the symptoms that afflict patients diagnosed with IBD are abdominal pain, arthralgias, fatigue, diarrhea, gastrointestinal bleeding, anorexia, weight loss, and nausea [1]. Treatment is aimed at relieving symptoms and trying to induce and maintain remission [2]. Because there is no cure and therapy has limited effectiveness, many patients turn to complementary and alternative medicine to achieve symptom control [3,4]. Among the reasons patients may have for seeking this type of therapy may be ineffectiveness of the treatment they are currently receiving, previous abdominal surgery, and chronic analgesic use, but further investigation is still needed [3,4,5]. Cannabis use, medicinal or recreational, is one of the options chosen by patients. There is little scientific data and no currently accepted medical indication for the use of cannabis as a treatment for IBD [6]. Despite this, its use has become frequent. In Puerto Rico, cannabis is legal for medicinal purposes. We defined medical cannabis or medical marijuana as cannabis used for medical purposes when it is bought legally and with a physician’s prescription.

The *Cannabis sativa* plant is composed of many chemical components, such as: terpenes, carbohydrates, fatty acids, esters, amides, amines, phytosterols, phenolic compounds, and cannabinoids. The two main cannabinoids are cannabidiol (CBD) and tetrahydrocannabinol (Δ⁹-THC). The main difference between them is the psychoactive component found only in THC, producing the common perception of “a high”. These compounds can activate endogenous receptors, namely, CB1 and CB2. All of these compounds and receptors make up what is known as the endocannabinoid system (ECS) [7].

The ECS is found in all vertebrates and humans and is distributed among organs and tissues. CB1 is mostly expressed in neurons of the central, peripheral, and enteric nervous systems, while CB2 is found mainly in immune cells. In the gastrointestinal system, CB1 and CB2 are found in all layers of intestinal sections. ECS components are involved in many gastrointestinal tract mechanisms. These mechanisms include the regulation of food intake, satiation, nausea, vomiting, gastric secretion, gastroprotection, gut motility, visceral sensation, intestinal inflammation, maintenance of epithelial barrier integrity, and immune tolerance regulation. It has been demonstrated that CB1 and CB2 receptor agonists reduce intestinal inflammation, especially CB2 [3,7].

Numerous studies have linked the ECS system to visceral pain perception, nausea, vomiting, gastrointestinal motility, and intestinal inflammation. The system’s multifaceted nature has led to the study of cannabis’s therapeutic use. Even though the United States Food and Drug Administration (FDA) has not approved cannabis plants for medical use, there are FDA-approved products that contain cannabinoids, such as cannabidiol (indicated for treatment of specific seizure disorders), nabilone (indicated for nausea associated with cancer chemotherapy), and dronabinol (indicated for anorexia and weight loss associated to AIDS and for chemotherapy-related nausea and vomiting) [8,9]. The pathogenesis of IBD is multifactorial, involving genetic susceptibility, altered immune response, external factors, and an abnormal intestinal microbiome characterized by dysbiosis. A recent systematic review and meta-analysis in the relation between cannabis and the microbiome offers evidence that cannabis may have an effect on the intestinal microbiome, increasing short chain fatty acid producing species and decreasing some pro-inflammatory ones. Cannabis has also been reported to have immunosuppressive characteristics. Early studies have shown that cannabinoids may alleviate intestinal inflammation in experimental IBD models. These properties of cannabis contribute to the interest in exploring a therapeutic role for it in IBD [7,10,11,12].

Cannabis use is not unchallenged from a health perspective. There are several long-term risks associated with cannabis use in the general population, including addiction, increase in motor vehicle accidents, symptoms of chronic bronchitis, abnormal brain development in a younger population, memory decline, reduced fertility, and psychiatric disturbances. Frequent cannabis use can also cause a syndrome of hyperemesis, and it increases the risks in surgery [13]. However, several studies conducted on patients with IBD have shown very little, if any, adverse effects to the use of cannabis. The most commonly reported effects were dry mouth, light-headedness, mobility/balance/coordination difficulties, somnolence, euphoria, and male impotence [14,15,16].

Apart from its innate effects, cannabis has the capacity to interact with the body’s main mechanism for pharmacological metabolism, the cytochrome P450 family of enzymes. These enzymes are responsible for the oxidation, reduction, and hydrolysis of many substances. Among these enzymes, cytochrome P450 (CYP) 3A4 is the most relevant determinant of drug metabolism and exposure for medications prescribed today. CYP3A4 is very important to intestinal drug metabolism. Cannabinoids have been proven to inhibit CYP2C19, CYP2C9, and CYP1A2. CYP3A4 may also be potentially inhibited, especially by CBD. Since most IBD medications are metabolized by these enzymes, concomitant use of cannabis may decrease their metabolism and allow an increase in their concentration. The metabolisms of azathioprine and methotrexate are decreased with cannabis use, while infliximab and adalimumab’s metabolism are increased. All of these potential adverse effects and interactions must be considered by the physician and the patient when using medical cannabis for IBD [17,18].

Recent studies in the form of surveys have been conducted to investigate the use of cannabis in patients with IBD. These studies aimed to protect the anonymity of patients while obtaining demographic characteristics, quality of life, patterns of use of cannabis, and the level of interest of patients in further cannabis research for use in the control of symptoms of IBD. Abdominal pain was the main symptom for which the patients were seeking relief in most studies [1,4,6,19,20]. Other symptoms were nausea, poor appetite, and diarrhea [1,2,6,19]. Previous studies have found that people diagnosed with IBD that used cannabis to aid in the relief of their symptoms were more likely to be male, aged between their 30 s and 40 s, and mainly used cannabis by smoking joints [2,3,9]. In another study, patients with IBD that used cannabis specifically for their symptoms were identified; some engaged in its recreational use, while others were prescribed medical marijuana for their IBD [6].

In 2014, the government of the Commonwealth of Puerto Rico (PR) legalized medicinal cannabis for several specific indications, including Crohn’s disease. The process does not require the recommendation of a gastroenterologist, and prescription privileges are limited to specific certified physicians. Patients with IBD and their relatives inquire about the benefits of the drug and an undetermined number are using medicinal cannabis, with or without their physicians’ knowledge and/or approval. Pronouncements by national gastroenterology professional associations and experts do not recommend the use of cannabis for the treatment of IBD in view of the lack of the literature supporting its benefits. Yet, a recent study of gastroenterologists in Puerto Rico revealed that physicians lacked information on the therapeutic benefits of cannabis and showed enthusiasm towards receiving more education on the topic [20].

Understanding the use of cannabis in patients with IBD can be helpful for physicians treating this population. Knowing the prevalence of cannabis use in these patients and the purpose and pattern of use could be useful in improving physician–patient communication, designing treatment strategies, and addressing patient needs. To attempt to answer some of these questions, we designed a cross-sectional study that collected information about cannabis use among patients attending the IBD clinic. The study will also help to create awareness among the physicians taking care of the IBD population and expand the current literature about cannabis use as an alternative for the symptomatic management of IBD.

## 2. Materials and Methods

Adult patients (aged 21 or older) with confirmed IBD attending the clinics at the University of Puerto Rico Center for Inflammatory Bowel Diseases were recruited for the study. All interested patients received an information sheet explaining the purpose and procedure of the study, together with an educational brochure about cannabis. Patients who accepted to participate were escorted to a private room in the IBD Clinics and, after oral consent, completed an anonymous questionnaire. With the intent of promoting participation and guaranteeing confidentiality, the questionnaire did not contain identifiers.

To increase the number of participants, we identified subjects from another study (the University of Puerto Rico Registry of IBD, Institutional Review Board protocol # 2290033794) who had agreed to be contacted about other research projects. These participants were recruited by the researchers via telephone and provided the option of online participation, using a compilation of the documents through a Google Form (Google Workspace™). The Google Form contained the study information sheet, the cannabis educational brochure, and the IRB-approved survey. Google Form provides for anonymous return of the survey. Figure 1 shows the study flowchart.

This is a sample of convenience based on patients visiting the IBD clinic. A target of 150 was chosen considering the number of unique patients in the clinics (around 900), the frequency of yearly visits (two to three), and the limitation in presential visits during the first year of the pandemic.

The questionnaire, designed by the investigators and shown in Figure 2, consists of 27 questions including demographics (age group and sex), clinical diagnosis (CD or UC), and use of cannabis. For respondents who did not report cannabis use, their participation in the questionnaire concluded in question #13. For participants who indicated that they were using or had used cannabis, specific questions followed regarding the mode of administration (inhaled, ingested, oil, pills, vaporized, topical application), frequency (many times per day, once a day, less than twice per week, at least twice per week), purpose (recreational or medicinal), source, cost of cannabis, and for former users the reason for discontinuing it. Participants were specifically asked to answer if they had used cannabis to alleviate IBD-related symptoms and if it had relieved their symptoms. The questionnaire can be completed in 15 min or less and does not include any identifiable information or date of completion. It is in Spanish, which is the native language of our patients. The questionnaire was evaluated by adult volunteers of both genders and a wide age range for clarity of language and interpretation. Any perceived difficulty was addressed by the investigators, and the modified questionnaire was again tested with volunteers. The questionnaire, translated into English, is shown in Figure 2. Exclusions to the study were unwillingness to participate, inability to fill out the questionnaire, being a minor under PR law, and not having a confirmed diagnosis of IBD. The protocol is approved by the Medical Sciences Campus Institutional Review Board (protocol #2290032828).

To establish a profile of the participants, a descriptive analysis employing frequencies, percentages, ranges, and measures of central tendency was performed with the obtained demographic data, IBD related details, and their past and present cannabis usage habits, further delving within those with reported consumption. To assess the differences between past and current users in profile, perceptions, and willingness to recommend cannabis, the unadjusted relationship of these variables was assessed through Fisher’s exact test for categorical variables, as well as through the two sample Wilcoxon-Mann-Whitney rank-sum tests for ordinal and continuous ones. A similar approach was used to explore possible differences between those that would ultimately recommend this medical alternative to others, as well as those that would not. For all tests, a *p*-value of 0.05 or less was considered an indicator of a significant difference. Statistical analysis was performed using Stata 14.2 (Statacorp LLC, College Station, TX, USA).

## 3. Results

An amount of 162 patients completed the survey, of which 37% (*n* = 60) were using (*n* = 35) or had used (*n* = 25) cannabis. Table 1 shows demographic characteristics of the participants of the study. Among cannabis users, most (38, 63.3%) were males; 45/58 (77.6%) had CD, and 39% were between 21 and 30 years old. An amount of 76.7% of the subjects who identified as cannabis users were 40 years old or younger; more than half had advanced schooling (college degree or beyond). Cigarette smoking was infrequent in our participants; only 7 of 60 (11.7%) cannabis users smoked cigarettes, while 5 of 102 (4.9%) non-users did. The participants had acquired information about cannabis from various sources, including a physician (43.4%), the internet (66.7%), social media (51.6%), and a relative or friend (73%). In spite of all the reported sources of information, most participants (77%) had low or moderate knowledge about cannabis, and 15% admitted having no knowledge. The most common methods of administration were inhalation (68%) and oral ingestion (58%), followed by vaping (43%) and oil ingestion (40%). Most (41.1%) used cannabis less than twice a week, 30.4% used it once daily, and only 16% used it more than once daily.

The cannabis users were asked their perception of disease severity at the onset of symptoms, and 38 (63.34%) graded it as severe or intolerable. However, most (43, 71.7%) perceived their disease as mild to moderate at the time of the study. Most used cannabis for medical reasons, 38 (63.3%) used it for relief of IBD, and most obtained it through a prescription (56.4%). Table 2 shows the symptoms for which the subjects used cannabis. Other symptoms include anxiety, insomnia, depression, and headache. Almost all (84% to 94%) reported an improvement of symptoms and quality of life, considered cannabis beneficial to their health and would recommend it to other patients with IBD. Among subjects that used cannabis for IBD, almost all (94.3%) have used it for more than a year and (34%) reported use of five or more years. The majority (43.6%) of users spend no more than $50 (USD) a month on their cannabis. Use of other analgesics was frequent. These included acetaminophen (45%), gabapentin (15%), tramadol (11.7%), corticosteroids (11.7%), opiates (8.3%), meperidine (6.7%), and others (1.7%).

An amount of 73.3% of cannabis users reported having discussed cannabis use with their physician, and a larger percentage (88.3%) of cannabis consumers said they would feel comfortable discussing medical cannabis for IBD with their physician. Among non-users, only 8% have discussed cannabis with their physician, but 96% said they would feel comfortable doing so. Only four participants out of 162 reported never hearing about medical cannabis. The most common reason for discontinuing the use of cannabis was having a job that did not allow cannabis use (36.0%), followed by not enjoying its effect (28%), absence of symptom relief (24.0%), and economic difficulties (8%). Other reasons included pregnancy, adverse interaction with other medications, having an expired cannabis license, or carrying a firearm permit. None referred to the COVID-19 pandemic as a cause for discontinuing the consumption of cannabis. Table 3 shows perceptions and attitudes of the cannabis consumers in the survey.

On bivariate analysis, current use of cannabis among all-time users did not vary significantly by age group, gender, or level of education. When comparing current with former users, there were no significant differences related to demographics, source of information, indication for use of cannabis, prescribed use, and frequency of use. Current users reported improvement of symptoms (*p* = 0.004) and benefit for their health (*p* = 0.02) significantly more than former users. However, there was no difference between groups in recommending cannabis to others with IBD. Age, gender, level of education, method of administration, and symptoms for using cannabis were not significantly associated with recommending cannabis use. However, using cannabis for IBD was significantly associated with recommending cannabis to other patients (*p* = 0.013), as was considering it beneficial to health (*p* = 0.008) and improvement in quality of life (*p* = 0.002)

## 4. Discussion

There are limited studies and data supporting the effectiveness of medical cannabis as a potential therapeutic aid for IBD. This study describes the current demographics and perception of patients with IBD in our clinic that have or have not used medical cannabis for their disease, with special focus on patients with reported use. After 162 participants were surveyed, results showed that over a third (37%) of our patients with IBD reports using medical cannabis, a higher prevalence than reported in other studies, including a previous one in our clinics [21,22]. A study in Canada in 2018 found a 54% rate of marijuana use for symptom control in IBD, further suggesting that patients are willing to explore the perceived benefits of cannabis [23]. In our study of 162 participants, we see a predominance of male patients with Crohn’s disease and of younger age between the ages of 21–30, in line with other studies in which male gender predominates among cannabis users [2].

The relief of specific symptoms described on Table 2 is especially important. The purpose of using cannabis in this population is to reduce their symptoms of IBD. Among the symptoms leading to cannabis use, abdominal pain, diarrhea, and loss of appetite are the most frequently reported in different studied populations, including ours. These studies are international, gathering a large sample size of different cultures and ethnicities. Most of the studies on cannabis and IBD report abdominal pain as the most frequently relieved symptom, as did our study. One of the first studies showing improvement with cannabis was performed in Israel. This was an observational study in a small population of patients with CD, of whom 21 out of 30 patients reported an improvement in their condition as measured by the Harvey-Bradshaw Index for disease activity [24]. Although no placebo control group was present in the study, it opened the door for other clinical studies regarding the potential benefits of medical cannabis in IBD patients. In our study’s population, most participants reported improvement in the symptoms, abdominal pain being the most frequent. The caveat of this finding is that the characterization of symptom improvement is entirely subjective and dependent on the patient’s perspective. Obtaining objective parameters to measure a reduction in inflammation would require measuring stool or blood biomarkers, such as C-reactive protein levels, erythrocyte sedimentation rate, and fecal calprotectin to monitor disease activity before and after cannabis consumption, which was beyond the scope of our study.

There was no specific demographic associated to discontinuation of cannabis by users, and lack of symptom relief was reported in only a fourth of cases. Most were driven by work restrictions, but whether these restrictions were task-related, general employer policies, or motivated by the U.S. federal government’s persistent prohibition of cannabis use was not explored. Should cannabis prove to be a valuable therapeutic option for IBD, this aspect will require further clarification.

There have been a few studies of primary care physicians and health professionals and their perception on medical cannabis, but little has been reported on patient’s perception and experience with medical marijuana. In our study, we aimed to explore this by adding questions in the survey such as what sources patients used for information on medical marijuana, how they obtained cannabis, if they had discussed with their treating physician the use of cannabis for IBD symptoms, and if they felt it had helped their symptoms. As the results show, a wide variety of sources played a role in informing patients about the use of medical marijuana in IBD treatment.

A study in 2016 [25] investigated the patient’s process to use medical marijuana, with a survey including questions on their experience, satisfaction, and motivation to use it. A specific question, “what prompted patients to seek treatment”, is similar to one in our study. The highest percentage (55%) reported it was their physician, followed by written information (15%), a friend (13%), and media (10%). When comparing our results with these findings, less than half of patients gained information from physicians (43.4%), but the internet (66.7%), social media (51.6%), and a relative or friend (73%) had a greater impact and were likely the main sources to prompt its use. Although many patients will research the topic on their own, the willingness of patients to try medical marijuana makes it important for physicians to be informed on the role of cannabis in the treatment of IBD in order to appropriately counsel their patients. A recent online survey given to gastroenterologists in Puerto Rico evaluated the readiness of these practitioners to discuss the use of medical cannabis with their patients. More than half of participants referred having no knowledge of the subject, although the majority were highly interested in learning more to counsel their patients [20].

The legal debate on cannabis is also a contributing factor to the outcomes of studies such as this one, since patients may be more willing to try cannabis in areas that have legalized its use and there is less social stigma attached. A recent analysis from the UK Medical Cannabis Registry [26] surveyed 76 patients with IBD, using four different objective scales for the effect of cannabis on health-related quality of life. The study showed statistically significant improvement after one and three months of follow-up, and most patients (almost 80%) reported no adverse effect. In this same study, the demographics of cannabis consumers showed a predominance of CD vs. UC. A large Australian study with similar inclusion criteria also showed few adverse effects after the use of cannabis, yet such recent studies only measure short-term benefit [27]. Both of these countries, the United Kingdom and Australia, have legalized and prescribe medical cannabis. We found no similar studies from a country where access to cannabis is illegal, probably because data is limited in these countries. Specific laws on the use of cannabis may affect patient perceptions as well as physician knowledge about its use for symptomatic treatment of disease.

One of the aims of our study was to explore patients’ understanding and perception of the use of cannabis. In this cohort, 94% of cannabis consumers considered cannabis to be beneficial for health, and 86% would recommend it. Few studies have approached this question on consumers’ perception of the subject. Although it is important to study the physiological effect of cannabis use on IBD, it is equally essential to understand patient perceptions so physicians can give adequate advice on an individual basis.

This study is part of an ongoing project to describe cannabis use in the IBD population. Data are being collected through the medical review of the medical record of the participants who consented including medications, past surgeries, and hospital stays. The medications of interest are the main IBD medications, any other concurrent medication, vitamins, supplements, or any other type of complementary medication, as documented in the medical chart. The many potential drug–drug and cannabis–drug interactions are outside of the scope of this exploratory descriptive study focusing mainly on the prevalence of cannabis use in our patients with IBD, but we believe they are important to consider. Of interest also would be the effects of previous surgeries and the differences between cannabis bought in the licensed dispensary or through illicit means.

There are several limitations to this study. The study has a small sample size, reflecting that recruitment was primarily based on attendance to the IBD clinics, and it does not necessarily reflect a lack of willingness to participate. Because we designed the study to provide full anonymity, a log of approached patients was not kept, so the response rate is unknown, and we are unable to compare it with other studies. Furthermore, avoiding duplication of subjects depended on the patient’s report of prior participation. This study did not ask for nor explore the different adverse effects from cannabis and its utilized dosage. The study also did not ask the participants about their medication use, use of supplements, antioxidants, or any other complementary medicine. It documented analgesics and pain reliever use in a limited way. We avoided free text answers in the survey because they prove difficult to analyze. Comorbid illnesses were not documented, but they could be added to the ongoing record review. It is important to note that we only reported symptom relief, but we did not explore actual therapeutic effects on the inflammatory process of IBD. Likewise, no validated instruments were used to explore effects on quality of life. Furthermore, medical cannabis is currently legal in Puerto Rico, yet the need for having a government-issued license may have influenced participants that acquire cannabis through other ways for symptoms related to IBD to not have answered truthfully.

## 5. Conclusions

The use of medical cannabis to relieve symptoms is frequent in patients with IBD, although knowledge about cannabis among patients and physicians is limited. In spite of symptom relief and perception of benefit, almost half of the users had stopped consuming cannabis. Our study supports the need for more investigation in this area, as well as an increase in educational programs for patients and physicians.

## Figures and Tables

**Figure 1 ijerph-20-05129-f001:**
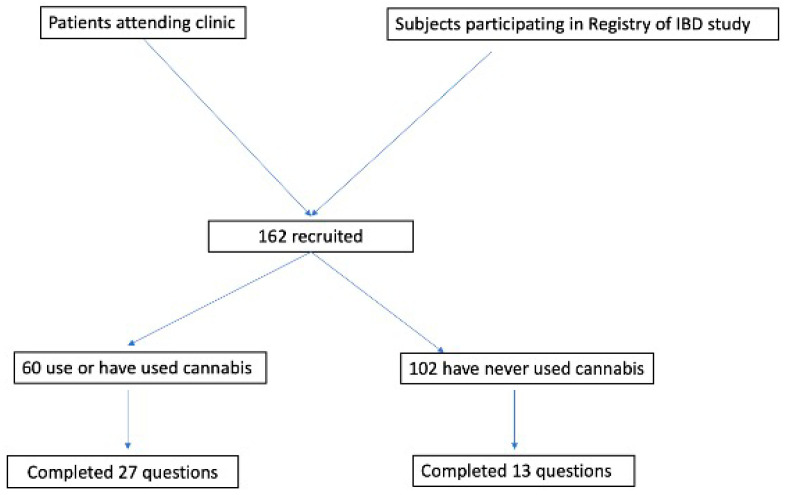
Flow chart of study.

**Figure 2 ijerph-20-05129-f002:**
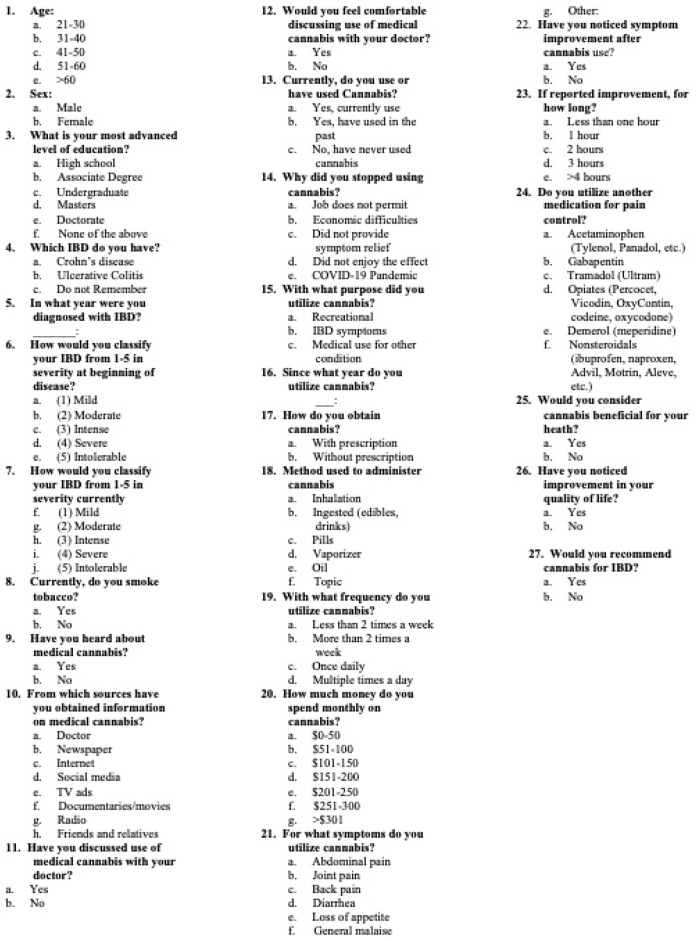
Questionnaire.

**Table 1 ijerph-20-05129-t001:** Demographics of all participants.

Cannabis Use (*n*, %)	Gender M:F	Mode Age Group (*n*, %)	CD:UC
Current or former (60, 37%)	38:22	21–30, 31–40 (46, 78)	45:13
Current (35, 58%)	22:13	21–30 (16, 46)	26:11
Former (25, 42%)	16:9	31–40 (11, 44)	21:5
Never (102, 63%)	47:55	21–30 (29, 28)	78:28
Total (162, 100%)	85:77	21–30 (53, 32.7)	123:41

Each mode of age groups calculated from *n* of categories of cannabis users in the same row in the first column of table. Abbreviations: M—male, F—female, CD—Crohn’s disease, UC—ulcerative colitis.

**Table 2 ijerph-20-05129-t002:** Symptoms for which cannabis was used.

Symptom	Frequency	Percentage
Abdominal Pain	39	65.0%
Joint Pain	23	38.3%
Back Pain	21	35.0%
Diarrhea	10	16.7%
Weight Loss	25	41.7%
Other	18	30.0%

**Table 3 ijerph-20-05129-t003:** Patient perceptions by gender on cannabis use for IBD as answered in survey (*n* = 60).

	Would Recommend Yes (%)	Improvement in Quality of Life Yes (%)	Beneficial for Health Yes (%)
Male	29 (82.9)	27 (80%),	31 (88.6)
Female	20 (90.9)	20 (90.9),	22 (100)
Total	49 (86)	47 (84),	47 (94)

## Data Availability

The data presented in this study are available on request from the corresponding author. The data are not publicly available due to ethical and privacy concerns. Participants recruited did not agree to sharing their data beyond the investigators, and this was specified for IRB approval.

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
