# Peer review of "A Survey of Cannabis Use among Patients with Inflammatory Bowel Disease (IBD)"

_ijerph, 2023, doi:10.3390/ijerph20065129_

Round 1

Reviewer 1 Report

The manuscript discusses an important issue which is in the scope of the journal. The manuscript aimed to survey cannabis use among patients with inflammatory bowel disease which is considered a very interesting topic to the public. Some clarifications as well as certain modifications are required to be performed through the following comments; 

1-      The mechanism of cannabis in relieving the symptoms of IBD, SHOULD be mentioned in the manuscript in full details including its pathway in this issue. Kindly add this information.

2-      Missing of question including the use of any complementary medicine by the patients. Clarify?

3-      Missing of question concerning the use of any supplementation of anti-inflammatory as well as antioxidants which might decrease the severity of associated symptoms with IBD. Clarify? otherwise to be added in the limitation section of the manuscript.

4-      The tolerability and safety of cannabis were missed. The authors SHOULD have added a question concerning this issue in order to get a complete information for the pros and cons of cannabis. This to be added in the limitation section.

5-      The validation of the questionnaire was missed. This is an important part for the reliability and reproducibility of the results. Please clarify why validation method was missed?

6-      Concurrent medications were missed in the questionnaire which is considered an important confounder ion the results concerning pain alleviation. Please clarify why this question was absent?, otherwise, this is to be added as a limitation to the study under limitation section.

7-      Question concerning the comorbidities of the patients was absent, this is an important information which will be accompanied by the treatment used for these comorbidities which might include drugs that either improve or deteriorate the IBD condition of the patient which might be misinterpreted that it is due to cannabis consumption. Please comment on this absence of such an important data and it SHOULD be added under section limitation in the manuscript.

8-      Dysbiosis which is one of the main reasons of digestive tract diseases and has to be mentioned as an important etiology and information as well to the readers. In the introduction the following paragraph is recommended to be added;

   Dysbiosis is mainly correlated with different GIT dis­ease conditions and dietary habits have an influence of normal state of commensal gut microbiome, and probiotics showed to have a significant effect in enhancing gut ecosystem, and support management of dif­ferent GIT disease conditions (1),

Reference

(1)Sara AR, Eslam MS, Raslan MA, Sabri NA. Correlation between Dysbiosis and Incidence of Gastrointestinal Cancers. Japanese J Gastroenterol Res. 2022; 2(11): 1102.

7-Study flow chart was missed. It is better to add study flow chart as possible clarifying and summarizing to the readers the pathway of the research.

8-The expected adverse effects as well as different drug - cannabis interaction was missed in the manuscript as an important scientific knowledge which might interfere with several treatment used in general, thus, updated studies performed on this topic are recommended to be cited in the introduction.

9- Page 6 line 271, what does this sentence mean “Only 7 cannabis users smoked cigarettes, while 5 of non-users did”. It seems that there is a missing word(s). Please revise it and correct.

10- Any abbreviation used in the Tables SHOULD be written in details at the footer of the tables.

11- What is the significance of the question and answer data represented in Table 2. Symptoms for which Cannabis was used.

12- Sample size calculation was missed. This SHOULD be presented with references and showing the power of study used as well as the full details of calculation.

13- What did the authors mean by the term used all over the manuscript: “medical cannabis”?

14- Different pathways as well as mechanism of action of cannabis in treatment of IBD are to be presented in the manuscript.

Author Response

Thank you for your comments and suggestions. We have addressed them in the revised manuscript. Regarding your specific concerns:

Question 1: Mechanism of cannabis was added to introduction

Questions 2 to 4, 6,7: We agree that information regarding other alternative and complimentary medications as well as other treatments for IBD is relevant to the subject. However, the design of the study was a simple, self-fill, multiple choice anonymous survey that did not lend itself to more details. We have an ongoing sub-study that includes reviewing the medical records of the participants with history of use of cannabis who gave consent to  access their charts. That review includes co-morbidities, all recorded medications, and surgery for IBD.  We added this as a study limitation in the discussion.

Question 5: The description of the questionnaire was expanded to include the pre-study evaluation of the investigator-created instrument. We should point out that it was created only in Spanish, which is the language of our country, and the English version included in the manuscript is simply a direct translation.

Second question 7: A flowchart of recruitment process was added as Figure 1. Questionnaire now becomes Fig 2.

Question 8 : The pathogenesis of IBD including dysbiosis was added in the introduction. The effects of cannabis as immunosppressant and on the microbiome relevant to IBD were also added , with additional references. We focused on IBD and not other conditions, as this the subject of our study.

Question 9: Sentence was rewritten and expanded for clarity.

Question 10. Abbreviations used in Tables were added as footnotes.

Question 11. Table 2 refers to the symptoms identifed by the participants as the ones for which cannabis was used. This was clarified in the text

Question 12. Sample size discussion was added to the Methods section.

Question 13: The definition of medical cannabis was added in the introduction, it means cannabis acquired legally with a prescrption for medical purposes, as is defined in Puerto Rico law.

Question 14: Properties of cannabis relevant to IBD were added to the manuscript.

Reviewer 2 Report

The present article written by Alondra Velez-Santiago and collegues is interesting and well written. However, here are some suggestions for improvement:

- Introduction - Line 102: Please write what IRB means. The same for Line 121 "MSC"

- The questionnaire could be put as an Appendix.

- Line 246, question 24. Please write first de common international name for all the medicines. Ex: for Demerol --- pethidine/meperidine. Same for nonsteroids.

- Could you please add to the results some information regarding the questions: 18 and 19 ? The method of use, as well as the frequency of cannabis utilization ? I do not recall seeing anything written in the manuscript.

- Please add an Abbreviation list at the end of the manuscript.

Author Response

Thank you for your comments and suggestions. We have revised the manuscript accordingly. 

  1. IRB and MSC have been spelled out.
  2. We agree the questionnaire added as Figure 1 could be an Appendix. We defer to the editors for that decision, as we feel they know best what serves the journal's style and audience best.
  3. The generic names for the medications were used in the results section. As the questionnaire was designed to be self-filled, we used brand names so participants could identify them. We have added the generic names to the translated questionnaire provided in the manuscript.
  4. Results for questions 18 and 19 were expanded in the Results section of the manuscript. They had been only briefly mentioned in the original.
  5.  An abbreviations list was added at the end of the manuscript.
  6. Additionally, the introduction has been expanded and more references have been added. Methods were detailed more, as were some of the results. The text has been edited for more clarity.